# From Species to Genes: A New Diagnostic Paradigm

**DOI:** 10.3390/antibiotics13070661

**Published:** 2024-07-17

**Authors:** Sinead Fahy, James A. O’Connor, Roy D. Sleator, Brigid Lucey

**Affiliations:** 1Department of Microbiology, Mercy University Hospital, T12 WE28 Cork, Ireland; sinead.fahy@mycit.ie; 2Department of Biological Sciences, Munster Technological University, T12 P928 Cork, Ireland; james.oconnor@mtu.ie (J.A.O.); brigid.lucey@mtu.ie (B.L.)

**Keywords:** taxonomy, molecular diagnostics, sequencing, mobile genetic elements, microbiology, element-borne genes

## Abstract

Molecular diagnostics has the potential to revolutionise the field of clinical microbiology. Microbial identification and nomenclature have, for too long, been restricted to phenotypic characterisation. However, this species-level view fails to wholly account for genetic heterogeneity, a result of lateral gene transfer, mediated primarily by mobile genetic elements. This genetic promiscuity has helped to drive virulence development, stress adaptation, and antimicrobial resistance in several important bacterial pathogens, complicating their detection and frustrating our ability to control them. We argue that, as clinical microbiologists at the front line, we must embrace the molecular technologies that allow us to focus specifically on the genetic elements that cause disease rather than the bacterial species that express them. This review focuses on the evolution of microbial taxonomy since the introduction of molecular sequencing, the role of mobile genetic elements in antimicrobial resistance, the current and emerging assays in clinical laboratories, and the comparison of phenotypic versus genotypic analyses. In essence, it is time now to refocus from species to genes as part of a new diagnostic paradigm.

## 1. Introduction

Traditionally, bacterial nomenclature was based exclusively at the genus and species levels [1], which were initially derived from visualised organism characteristics, and phenotypic characterisation [2]. Today, techniques such as genomic sequencing are helping refocus our view of taxonomic discovery and reclassification [3]. While more accurate, and nuanced, than traditional phenotype-based classification methods, molecular approaches to revising bacterial nomenclature are likely to have a significant impact on the field of clinical microbiology, where accurate pathogen identification is essential for fast and effective control [3]. This is particularly important against the current backdrop of increasing antimicrobial resistance [4]. In 2019 alone, for example, there were an estimated 1.27 million deaths attributable to Antimicrobial Resistance (AMR), globally [4]; these were driven by the indiscriminate use of antimicrobials, along with substandard healthcare, and agricultural antibiotic misuse [5]. 

The necessarily accelerated advancement of molecular testing, in clinical laboratories, driven by the COVID-19 pandemic, has resulted in a transformation in the way clinicians view and utilise diagnostic molecular testing, as shown by Afzal [6]. Our dependence on first identifying a set of isolates from normally non-sterile clinical samples, such as the respiratory or enteric tracts, suggests that taxonomy (which is dynamic, as shown by the membership of the family Enterobacterales over the past 40 years) [7] is the first step in identifying a pathogen. However, this approach has two potential problems, the first being that isolation attempts may be flawed and the second being that the dependence on taxonomy rather than virulence factors may sometimes subvert diagnosis [7]. 

Herein, we discuss the identification and treatment of bacterial infections in the molecular era. We highlight the contribution of mobile genetic elements (MGEs) to AMR and assess the impact of modern molecular-based detection methods, such as whole-genome sequencing (WGS), in clinical practice. Based on the above, we propose that there exists a real and immediate need to change from an identification strategy rooted in traditional phenotype-based taxonomy to one that is based on genetic identification, i.e., a refocusing from species-based identification to gene-based detection. 

## 2. Taxonomy and Clinical Microbiology

Taxonomy divides living organisms into three distinct categories, namely classification, nomenclature, and identification [3]. The current bacterial nomenclature landscape is complex, with more detailed phylogenetic analysis and the expansion of next-generation sequencing technologies leading to the identification of new species, as well as the re-classification of previously defined genera [8]. While many of these organisms appear to play a commensal role in site-specific ecology, some may be of clinical significance in certain patient populations [8]. 

For instance, *Campylobacter ureolyticus* (formerly *Bacteroides ureolyticus*) is a phenotypically distinct species that had previously been missed by traditional culture techniques but which we identified using the EntericBio^®®^ multiplex polymerase chain reaction (PCR) system, (Serosep Limited, Limerick, Ireland). Indeed, we have shown that this previously overlooked species may represent an important gastrointestinal pathogen [9,10]. Genotypically, this pathogen has significant implications for clinical patient care, antimicrobial susceptibility testing, data interpretation, or, indeed, the clinical relevance or epidemiology of the presumptive pathogen. 

Another example of such a taxonomic reclassification, and its potential impact within the clinical laboratory, specifically in the context of antibiotic prescription, is the renaming of *Enterobacter aerogenes* (part of the *E. cloacae complex* (*Ecc*)) as *Klebsiella aerogenes* [11]. A comparative study of clinical characteristics, outcomes, and bacterial genetics amongst a cohort of patients with *K. aerogenes* versus *Ecc* blood stream infections (BSIs) identified significant genetic differences previously obscured by poor taxonomy. Rates of resistance to trimethoprim–sulfamethoxazole (19% vs. 0%) and gentamicin (10% vs. 0%) were higher in *Ecc* isolates relative to *K. aerogenes.* WGS and pan-genome analysis conducted on 150 clinical isolates revealed 983 genes in 323 genomic islands unique to *K. aerogenes*; antibiotic resistance genes were largely found in *Ecc*, which also had a higher rate of multidrug-resistant (MDR) phenotypes [12]. Penicillin binding protein 3 (PBP3) was the most commonly present resistance gene, being found in 94% (141/150) of *K. aerogenes* and *Ecc* isolates [12]. This heterogeneity, particularly in terms of antibiotic resistance profiles, would have been unwittingly masked by the umbrella term *Ecc*—a fact that could potentially have led to suboptimal antimicrobial prescription and, consequently, diminished clinical outcomes. 

Examples of taxonomic rearrangement can be seen in the case of *enterobacterales* [13], in particular *Salmonella* and *Escherichia coli*, where hundreds of different serovariants may be differentiated based on the cell wall, e.g., O; somatic antigen, K; capsule or H; and flagellar antigens [14,15]. Similarly, bacteria may also be characterised, based on their disease-causing capacities, into pathotypes that include extraintestinal–pathogenic *Escherichia coli* (ExPEC), enterotoxin-producing *E. coli* (ETEC) or enteroinvasive *E. coli* (EIEC), and so on [16]. Using EIEC as an example of pathogenicity, it is closely related to *Shigella*, with a shared virulence determinant on plasmid pINV, which encodes a type-3 secretion system for the movement of bacteria into epithelial cells of the intestine [17]. EIEC and *Shigella* exemplify the changes *E. coli* is capable of undergoing in order to adjust to a pathogenic lifestyle. Indeed, the capacity of *E. coli* to survive and thrive in certain environments is enhanced by genomic modifications and mediated by the acquisition, deletion, or inactivation of genes [17]. This further highlights the need for diagnosticians to shift their focus from species to genes [16]. With the future of clinical microbiology lying in a more patient-personalised sequence profile, it may be more judicious to focus on genes, as virulence factors and markers of disease, than the bacterial species that express them, particularly given the degree of horizontal gene transfer that exists in ‘hot spot’ environments such as hospitals.

A virulence gene is defined as any gene that can increase pathogen fitness with the consequence of causing disease in either a qualitative or quantitative sense, including the adaptation of pathogens to host immunity, or the environment [18]. The distribution of virulence genes was investigated in 91 isolates of *Providencia* by Yuan et al. [19] using the Virulence Factors Database (https://ngdc.cncb.ac.cn/databasecommons/database/id/516, accessed on 4 July 2024). The genes *mgtB* and *mgtC*, which encode Mg2+ uptake in *Salmonella enterica* serovar *typhimurium*, were distributed in almost all Providencia strains [19]. MgtB is responsible for transporting Mg2+ from the periplasm to the cytoplasm in *Salmonella enterica*, with MgtC being required for survival inside macrophages [20]. In addition, genes encoding Type 1 fimbriae of uropathogenic *E. coli* 536 were mainly found in *Providencia rettgeri*. Type 1 fimbriae are the most common adhesins among both commensal and pathogenic isolates of *E. coli*, with their expression being linked to the successful establishment of urinary tract infections (UTIs) [21]. This research highlights the differential distribution of virulence-related gene clusters, explaining the differences in the pathogenicity of *Providencia* isolates. 

## 3. Mobile Genetic Elements and Antimicrobial Resistance Genes

The World Health Organisation’s (WHO) critical priority list for the development of novel antimicrobial agents is solely composed of MDR Gram-negative bacilli (GNB), namely *Acinetobacter* spp., *Pseudomonas* spp., and *Enterobacterales* [22]. Understanding the emergence and rapid expansion of clones, many of which harbour AMR determinants that are encoded by MGEs, is particularly important. Examples include *Klebsiella pneumoniae* carbapenemase (KPC)-producing *Klebsiella pneumoniae*, CTX-M-15-producing *E. coli*, and OXA-23-producing *Acinetobacter baumannii* [23,24]. Clinically relevant antimicrobial resistant genes are often encoded on plasmids or transposons [25]; a compilation of commonly occurring resistance genes can be found in Table 1. The acquisition of MGEs and expansion of MDR clones among GNBs is evolving; their integration into the bacterial genomes of different genera and species and the spread of resistant clones require significant further investigation. This highlights the need for the application of genomic tools to identify MGEs and elucidate the phylogenetic relationships that exist between the genomes of clinical isolates and their associated extrachromosomal elements.

Insertion sequences (ISs) and transposons are small MGEs; ISs can move resistance genes as part of a composite transposon, a region bounded by two copies of the same IS that can move as a single unit. Many include a strong promoter that drives the expression of the captured gene [26]. IS*1999*, for example, is an IS that encodes *bla*OXA-48-like carbapenem resistance [25].

Plasmids act as vehicles for the mobilisation of other MGEs and acquired antimicrobial resistance genes and can vary in size from less than a kilobase to several megabases [27]. Plasmids promote adaptability in several ways. First, in many cases, when a gene moves onto a plasmid, its copy number per bacterium rises; thus, the overall mutation rate is increased. Secondly, they are often self-transmissible or mobilisable so that they increase the chance of the gene moving between bacteria by horizontal gene transfer [28]. Thirdly, they remove the need for the gene to integrate into the bacterial chromosome in order to become established in a new bacterium [29]. The genes encoding these functions form a core of plasmid housekeeping genes that may be adapted to benefit the host cell in a particular environment [29]. These accessory regions are typically made up of resistance genes and associated MGEs such as insertions or transposons [25]. F plasmids were among the earliest to be associated with antibiotic resistance and appear to be the most abundant plasmid type found in *Enterobacterales.* F plasmids often carry a blaCTX-M gene, e.g., *bla*CTX-M-15 in *E. coli* ST131, which likely contributes to the success of this ST [30]. Fıı_k_ plasmids are associated with *bla*KPC in ST258. F plasmids carrying *mcr-1* have also been reported [31]. Plasmid classification commonly relies on the phenomenon of incompatibility in that closely related plasmids cannot coexist stably in the same cell. As molecular typing becomes more readily available, replicon typing and plasmid mobility (MOB typing) can be exploited to classify plasmids by their phylogenetic relatedness [32]. Large datasets can be analysed using in silico plasmid typing methods from WGS. However, there are difficulties in reconstructing plasmid sequences from short reads (100–300 bp), limiting the epidemiological insight [32]. Isolating individual plasmids prior to sequencing simplifies assembly, potentially enabling complete plasmid reconstruction, but is a laborious process [33]. As long-read sequencing becomes more cost-efficient, resolving accurate plasmid structures is the primary goal [34]. 

AMR genes are acquired, expressed, disseminated, and traded mainly by integrons [35] by transferring genes from bacterial chromosomes to plasmids. They consist of three essential core features; the first component of the integron is a gene that produces integrase, encoded by the *intI* gene, which is required for site-specific recombination inside the integron. The second component is an adjacent recombination site (*attI*), which the integrase recognises. The third component is a promoter (Pc), which is situated upstream and is required for the effective process of transcription and expression [36]. Gene cassettes are small movable components carrying a single gene, typically without a promoter or recombination site (*attC*). They generally lack promoters despite having a coding sequence, which acts as the system’s mobile component, and most cassettes encode resistance to a wide variety of antibiotics [35]. The integration of circular gene cassettes by C1 integrons occurs by site-specific recombination between *attI* and *attC*, assisted by the integron integrase [35]. According to previous studies, gene cassettes are randomly combined in the region between the 3′ and 5′ conserved segments of integrons; the process is reversible and cassettes can be released in the form of free DNA from integrons [37]. Integrons are immobile and are mostly found on transferable plasmids. These moving plasmids carry gene cassettes that can transfer to other integrons or even to the host bacterial genome. The integron system allows microorganisms to combine gene cassettes and convert them to functional proteins. MGEs containing plasmids, transposons, and genetic islands can act as reservoirs of information for integrons to be shared amongst bacteria [37]. *Klebisella* spp. are associated with pneumonia and urinary tract and blood infections. Although *Klebsiella* species are associated with many resistant genes, examples of integron-associated AMR include class-1 integron-associated Metallo-beta-lactamase (MBL) genes including verona integrin (VIM), Klebsiella pneumoniae (KPC), and imipenemase (IMP)-type carbapenemases [37,38].

MGEs, as described above, are known contributors to bacterial adaptation and evolution; however, high-throughput, unbiased MGE detection remains challenging [39]. A study carried out by Durrant et al. [39] described a bioinformatic toolbox known as MGEfinder, which uses short-read-sequence data to identify integrative MGEs and their insertion sites without the need for a complete genome assembly or a database of known elements. MGEfinder has the ability to identify MGEs along with their insertion sequences that are repeatedly ‘hit’ by insertional mutagenesis, such as acrR, a gene involved in sensitivity to many antibiotics [40]. The insertional loss of function of acrR is identified at a high rate in *E. coli*, *K. pneumonia*, and *A. baumanni*, which likely correlates with the increased antibiotic resistance of these organisms [39]. Although there is much work to do before MGE sequence types can become a reliable target for typing, it is anticipated that applying the MGEfinder workflow to a wide variety of bacterial species will greatly enhance our understanding of MGEs and their role in bacterial adaptation [39]. 

Evidently, there exists a multitude of putative resistance genes in the environment; however, we cannot predict which ones may be expressed in pathogenic bacteria or whether they will result clinical treatment complications [41]. With the application of sequence typing, understanding the factors that contribute to the spread of AMR will allow for more targeted antimicrobial treatment. 

## 4. Molecular Detection of Element-Borne Genes in Clinical Practice

GenBank, a genetic sequence database, can be used to track element-borne genes (genes carried by elements such as plasmids or transposons) across various genera and species and their associations with different microorganisms [42]. These genes, often found on MGEs like plasmids, can move within and between species, influencing the spread of antibiotic resistance genes (ARGs) [43]. The transferability of these elements plays a crucial role in the dissemination of resistance genes among different bacterial species and ecosystems [44]. Plasmids, as carriers of resistance genes, are significant in spreading antibiotic resistance, highlighting the importance of understanding their role in resistance dissemination [30]. 

Rapid molecular diagnostics have been discussed in several excellent reviews [45,46]. They present progress in Nucleic Acid Amplification Technology (NAAT), microarrays, and mass spectrometry applications but also highlight that very few have attained Food and Drug Administration (FDA) approval. The overwhelming variety of antimicrobials and resistance mechanisms complicates antimicrobial sensitivity testing (AST). Genotypic (nucleic-acid-based) methods can only find resistances that are searched for, and the potentially found resistance genes are not necessarily from the actual pathogenic organism [46]. Quick identification can efficiently restrict the search palette for certain antibiotics. Hence, mass spectrometry has become a versatile workhorse in clinical microbiology that is routinely applied for bacterial ID. Through the simultaneous measurement of several metabolites, a biochemical signature can be obtained. Matrix-assisted laser desorption/ionisation time of flight (MALDI-TOF) applies laser energy to evaporate the matrix-bound sample that is then immediately analysed. When frequent sampling is applied, MALDI-TOF can even provide semi-quantitative growth rate data. Bruker Corp. (Bremen, Germany) has launched test kits such as the BT STAR-Carba Assay for AST based on antibiotic degradation monitoring [47]. Mass spectrometry is likely to become tightly integrated into other AST technologies, especially in the diagnostics of critical samples such as blood cultures. Using MALDI-TOF Direct-On-Target Microdroplet Growth Assay (DOT-MGA), sample droplets (culture plus antibiotics in 6 µL volume) are spotted directly onto disposable MS-target plates, incubated for 3–4 h, and analysed with MS [48]. 

Quantitative PCR can provide the early detection of pathogens and resistance genes [49]. NAAT and hybridization techniques combined have led to the FDA approving multiplexed diagnostic panels, with Xpert^®®^ Carba-R as an example, (Cepheid, Espoo, Finland). This technology combines sample preparation, real-time PCR, and nucleic acid analysis with molecular beacons to identify AMR resistance in clinical laboratories [46]. The fast analysis of PCR products enhances the throughput of NAAT systems. T2Biosystems has recently launched a test panel capable of detecting 13 resistance genes from both Gram-positive and Gram-negative pathogens directly from blood. The amplification products are detected by magnetic resonance after hybridisation with DNA probes conjugated with superparamagnetic particles [50]. Older technologies such as nucleic-acid-sequence-based amplification (NASBA) have simplified molecular diagnostics, making them more robust, enabling miniaturisation, and reducing the costs of instrumentation by allowing NAAT at a constant isothermal temperature. NucliSENS R EASYQ (bioMerieux) was the first automated system to combine NASBA and real-time detection using molecular beacon probes. It enabled the fast detection of KPC genes in 111 isolates of 300 *enterobacterales* isolates that were harbouring KPC genes, with no false positives, and results were available within 2 h [51]. Genefluidics, another hybridisation assay, announced CE-IVD marking for the UtiMax^TM^ kit, (GeneFluidics, Duarte, CA, USA), which provides ID in 30 min and AST within 2 h directly from urine with an overall sensitivity of 100% and specificity of 98.2% [46]. 

By detecting resistance genes first, healthcare providers could anticipate resistance patterns and tailor treatment plans accordingly. This proactive approach could enhance patient outcomes by reducing the risk of treatment failure and the spread of resistant strains. One example is the prediction of antimicrobial resistance in *Pseudomonas aeruginosa* with machine-learning-enabled molecular diagnostics. A study conducted by Khaledi et al. (2020) sequenced genomes and transcriptomes of 414 drug-resistant clinical *P. aeruginosa* isolates [49]. The presence/absence of genes, sequence variation, and expression profiles generated predictive models with biomarkers of resistance to four commonly administered antimicrobials. Gene expression information improved diagnostic performance for three out of four antibiotics [49]. The implementation of such a molecular susceptibility test system in clinical microbiology settings has the potential to provide earlier and more detailed information on the antibiotic resistance profiles of bacterial pathogens and potentially revolutionise how clinical microbiology teams identify and treat bacterial infections.

## 5. Genetic Markers vs. Microbial Identification

Traditional phenotypic AST takes, on average, 2–3 days [52]. Clinical microbiology staff are competent in phenotypic AST testing, with well-established quality systems in place [53]. Phenotypic testing using a combination of organism identification and antibiograms (manual or automated testing) is performed on pure, cultured isolates that require time for the reporting of actionable results. This time-limiting step poses a significant problem for clinicians, particularly in emergency cases such as sepsis, where speedy intervention is essential [54]. AMR surveillance, led by national and international bodies such as the European Antimicrobial Resistance Surveillance Network (EARS-Net) and WHO, is based primarily on phenotypic results. 

In recent times, the development of specialised molecular diagnostic panels by commercial companies has had a significant impact in the clinical laboratory by simultaneously detecting organisms and AMR genes directly from clinical specimens, with the presence/absence of an AMR marker being used to predict phenotypic AST results [52]. There are a number of tests that can detect organisms and/or AMR genes from isolates after growth (e.g., solid media) or directly from specimens (e.g., nasopharyngeal samples) [55], offering rapid turnaround times (TATs) and improved prognostic outcomes, particularly in patients with BSIs in guiding antimicrobial therapy [41]. An example of one molecular assay currently in clinical use is the BIOFIRE^®®^ Blood Culture Identification 2 (BCID2) panel (BioFire Diagnostics, bioMerieux, Salt Lake City, UT, USA). This is a second-generation multiplex PCR system that is capable of detecting forty-three targets, including twenty-six bacterial, seven yeast, and ten antibiotic resistance genes, providing results within 1 h directly from positive-flagged blood culture bottles [56]. A retrospective observational study conducted by Chen et al. [57] involved 129 positive blood cultures from intensive care unit (ICU) patients who underwent BCID2 testing; its concordance with conventional culture methods and its impact on antimicrobial stewardship were examined [57]. The time from culture to obtaining BCID2 results was significantly shorter than for conventional methods (46.2 h vs. 86.9 h); BCID2 also demonstrated 100% concordance in genotype–phenotype correlation in AMR reporting [57]. A total of 40.5% of patients included in this study received inadequate empirical antimicrobial treatment, which was adjusted or confirmed in 55.4% of patients following BCID2 results, having a noticeable impact on antimicrobial stewardship [57]. 

Bacterial clones are defined as isolates that are indistinguishable, or highly similar, when identified using molecular typing [33]. Certain sequence types are associated with the carriage of specific AMR genes; the ability to differentiate infections caused by high-risk clones offers a significant advantage to clinical microbiology teams [33]. Eminent bacterial clones constitute a powerful source for the propagation of antimicrobial-resistant genetic components, i.e., integrons, plasmids, and transposons. They provide stable platforms for the maintenance and spread of genes responsible for the global emergence of multidrug-resistant *Enterobacterales* [58]. The outbreaks caused by MDR *E. coli* and *K. pneumoniae*, including cephalosporin- and carbapenem-resistant isolates, are due to the dissemination of certain high-risk clones, namely *E. coli* sequence type 131 (ST131) and *K. pneumoniae* (ST258). Both named clones have a strong affiliation with broad host range plasmids, IncF, containing FIA and FII replicon types. The reason for the success of these clones is uncertain, but their ability to spread swiftly is beyond dispute [33]. The identification of clones is dependent on the molecular typing method used; the most common methods currently in use include multi-locus sequence typing (MLST), pulsed-field gel electrophoresis (PFGE), and PCR typing. High-risk clones have acquired certain adaptive traits that increase their pathogenicity and survival skills, which are accompanied by the acquisition of antibiotic resistance determinants. These clones have the tenacity and flexibility to accumulate and then provide resistance and virulence genes to other isolates. High-risk clones have contributed to the spread of global multidrug resistance through the transmission of different types of genetic platforms, including plasmids, and resistance genes among Gram-negative bacteria [33]. Some rapid detection methods have been designed for surveillance studies on ST131 and ST258, with PCR being the most cost-effective, particularly for large numbers [33]. Next-generation sequencing (NGS) uses PCR to amplify individual DNA molecules that are immobilised on a solid surface, enabling molecules to be sequenced in parallel, leading to decreased costs and rapid turnaround times [59]. Several *E. coli* ST131 and *K. pneumoniae ST258* have undergone NGS, as described in a study of a KPC outbreak in an Italian hospital that was driven by an ST131 cluster [60]. It is likely that this technique, combined with more user-friendly bioinformatics, will become the gold standard for the identification of sequence types in the near future [60]. 

Many antibiotic-resistant pathogens are associated with higher transmission rates. A recent study on >1700 carbapenem-susceptible and -resistant *K. pneumoniae* sequences concluded that the degree of resistance is correlated with the spread and that carbapenemase-positive isolates had the highest rate of transmissibility, in particular the ST258 lineage [61]. 

Significant recent advances in molecular diagnostics, and the introduction of commercially available tests [62], have meant that traditional culture-based diagnostic techniques for gastrointestinal pathogens, in particular, are steadily being replaced by newer rapid antigen detection and molecular-based methods. Multiplex molecular assays are helpful, from a therapeutic point of view, to avoid inappropriate and unnecessary antimicrobial treatment. This is particularly important, for example, in the case of Shiga toxin-producing *E. coli* (STEC) infection, where antimicrobial exposure may increase the risk of patients developing haemolytic uremic syndrome (HUS) [62]. 

However, despite the obvious benefits of these molecular assays, several limitations need to be considered before their streamlined introduction into clinical practice. Firstly, the substantial capital investment and associated set-up costs of introducing molecular techniques to clinical laboratories need to be considered, particularly in the context of WGS [53]. A PacBio instrument alone reportedly costs in the region of USD 779,000 [63], not considering the associated bioinformatic analysis costs or specialist staff training required to run the instrument and interpret the sequence data. A further limitation is the constant need for change, with the diversity of targets included in broad-spectrum in-house PCR assays needing to be expanded, on an almost constant basis, to keep pace with emerging resistance [54]. Furthermore, bioinformatics and systems biology do not routinely form part of the core training in a typical clinical microbiology setting. Thus, to compensate for this, significant investment is needed in staff training and the introduction of standard operating procedures (SOPs) to ensure that molecular analysis can be exploited to the fullest extent [53]. Illustrating this need for increased training and standardisation, a recent inter-laboratory study reported discordant AMR predictions by WGS bioinformatics. The study found that participants predicted different numbers of AMR-associated genes and different gene variants from the same clinical samples [64]. Participants used a combination of methods to analyse their samples. SPAdes was the most common program used to assemble raw reads. For the identification of AMR-associated genes, ABRicate (https://github.com/tseemann/abricate, accessed on 5 July 2024) and Resistance Gene Identifier (RGI) were the most popular tools used, with both taking assembled reads as input [65]. One or a combination of AMR databases such as the Comprehensive Antibiotic Resistance Database (CARD), Resfinder, and Arg-annot were used to analyse samples [64]. Disparities were also reported based on the choice of database used to compare sequences [64]. Labs that repeatedly reported the highest number of genes per sample used the CARD consistently; this was due to the CARD including many sequences from loosely AMR-associated efflux pump genes that are not found on other databases [64]. Sequence identity and the breadth of coverage thresholds were also factors in the number of AMR-associated genes reported. An increased number of ‘hits’ were inferred when the lowest identity and breath coverage threshold were used [64]. Removing the discrepancies between databases used, a pairwise comparison between all participants found that two sets of participants only reported the exact same genes within a sample in 2% (18/900) of cases [64]. There were clear examples where participants assigned different gene variants to the same sequence data wherein the reference sequence differed only by a few single nucleotides [64]. It was also suggested that sequencing depth, resequencing, and small increases in sequence length can lead to variations in results [64]. Had the results been used to predict AST and guide treatment, a different antibiotic would have been recommended for each isolate by at least one participant [64]. Additionally, WGS quality control may be substandard, often resulting in false positives or false negatives if not complemented with phenotypic AST. For example, a result indicating a resistant phenotype and a positive molecular result implies the expression of the AMR gene; however, target loci may be present but not expressed, which can falsely predict AMR if a molecular test is used in isolation, which in turn may lead to a falsely elevated AMR reporting. Furthermore, failure to detect the presence of a new unknown AMR gene may result in a false prediction of the absence of AMR [66]. 

However, there is a fundamental need to identify both pathogens and AMR genes, both established and emerging, that impose the highest clinical and economic costs. Thus, the source of infection can be identified and effective infection control measures be implemented swiftly enough to be of use [67]. None of the above issues are insurmountable, and information sharing between different clinical disciplines may help ease the burden. For example, best-practice guidelines, by an expert group, for WGS in cancer testing have made recommendations on development, optimisation, and validation, including reference materials for the evaluation of assay performance and requirements for minimum sample numbers to be used when establishing test performance [68]. The recommendations are intended to assist clinical laboratories with the validation and on-going monitoring of WGS testing for the detection of somatic variants and to ensure the high quality of results [68]. Borrowing and adapting these oncology-based guidelines would likely have a positive impact on the field of clinical microbiology.

A significant goal for personalised medicine is permitting a shift from disease treatment to prevention. Pharmacogenomics, a promising area in precision medicine, can use molecular markers to assess drug efficacy, safety, and disease risk. For example, a recent study into co-trimoxazole, commonly used in the treatment of pneumonia, and the genetic predisposition to severe cutaneous adverse reaction (SCAR) used WGS to identify a single nucleotide polymorphism, rs41554616, having the strongest association with co-trimoxazole induced SCAR in Asians [69]. Although precision medicine still requires extensive validation before clinical uptake, with Figure 1 describing a potential workflow, it is a promising example in which molecular diagnostics and medicine can work side by side to improve patient outcomes. 

## 6. Conclusions

Although there are many factors to be considered before the streamlined introduction of molecular analysis, such as the substantial set-up cost, staff expertise, and the growth of an extensive database, the precise targeting of antibiotic resistance genes and other virulence-associated loci is likely to have a significant impact on the detection and prevention of recalcitrant pathogens in healthcare settings. Both established and emerging assays described in this review are proving to reduce TATs, improve antibiotic stewardship, and increase our reference databases for future genome analysis. Not only does personalised medicine revolutionise how infections are diagnosed, but this refocusing from bacterial species to their genetic loci, we feel, will also aid in the global fight against antibiotic resistance. Molecular analysis enables us to further our understanding of bacterial evolution and the ability of bacteria to carry, transfer, and express AMR genes via MGEs. Phenotypic discordancy between AMR needs further research before surveillance authorities can report on genotypes in clinical practice. The future of clinical microbiology is exponentially evolving to meet the demands of antimicrobial resistance; although these advancements are more prominent in developed countries, extensive analysis in economically underdeveloped regions needs to be upscaled as these areas typically account for increased levels of AMR. 

## Figures and Tables

**Figure 1 antibiotics-13-00661-f001:**
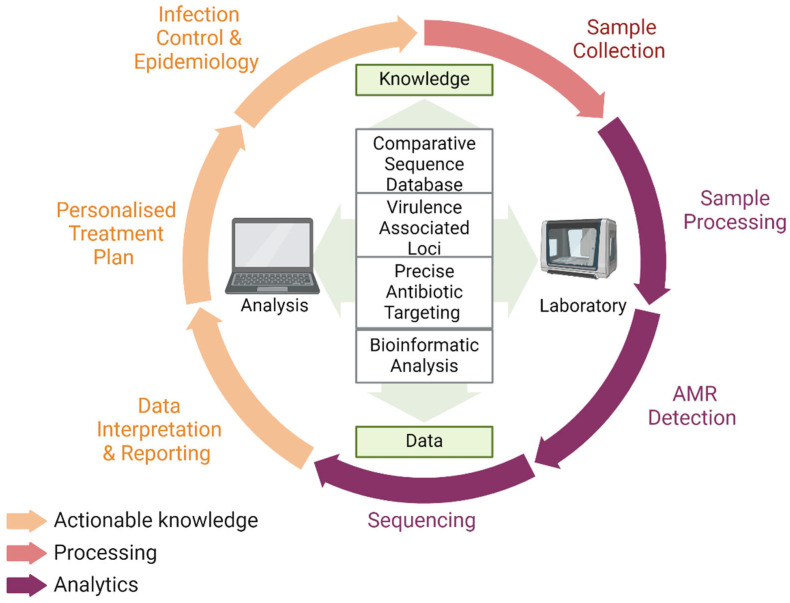
Potential workflow for AMR detection in clinical microbiology. Created with BioRender.com.

**Table 1 antibiotics-13-00661-t001:** Resistance genes found on mobile genetic elements in Gram-negative bacilli.

Resistance Gene	Antibiotic Class	Mobile Genetic Element	Mechanism of Resistance
*bla*CTX-M	β-lactams (Cephalosporins)	Plasmids, integrons	Extended-spectrum β-lactamase production
*bla*KPC	β-lactams (Carbapenems)	Plasmids, transposons	Carbapenemase production
*bla*NDM	β-lactams (Carbapenems)	Plasmids, integrons	Metallo-β-lactamase production
*bla*OXA-48	β-lactams (Carbapenems)	Plasmids, integrons	Carbapenemase production
*bla*VIM	β-lactams (Carbapenems)	Plasmids, integrons, transposons	Metallo-β-lactamase production
*qnr*	Fluoroquinolones	Plasmids	Plasmid-mediated quinolone resistance (PMQR)
*mcr-1*	Polymyxins (Colistin)	Plasmids	Phosphoethanolamine transferase production
*tet*(A)	Tetracyclines	Plasmids, transposons	Tetracycline efflux pump
*dfrA*	Trimethoprim	Plasmids, integrons	Dihydrofolate reductase alteration
*armA*	Aminoglycosides	Plasmids	Methyltransferase production (16S rRNA methylation)

## Data Availability

No new data were created or analyzed in this study. Data sharing is not applicable to this article.

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
