# Peer review of "From Species to Genes: A New Diagnostic Paradigm"

_antibiotics, 2024, doi:10.3390/antibiotics13070661_

Round 1
Reviewer 1 Report
Comments and Suggestions for Authors
Comments
1. Acronyms must first be defined as they appear in the text.
2. Authors must include one or two tables highlighting relevant information in the manuscript.
3. The authors must include a conclusions section.
4. The authors must include a figure that supports the conclusions of the review or manuscript.
Line by line Comments
Lines 4 – 6. The adscriptions are not complete
Lines 7 – 16. There is no good approach that emphasizes the objective of the work. The authors only mention that it is very complicated, frustrating, and requires other skills, but do not give examples to refer to the enunciation.
Lines 20 – 47. The introduction is well planned and where they want to lead the review.
Lines 49 – 77. There is a disorder when they describe the phenotypic and genotype part; that is, they describe phenotypic by genotypic, then return to phenotypic again.
Lines 78 to 94. The authors describe several examples of the diversities of some pathogens of interest; it is suggested that the identification extends to other genes such virulence genes.
Lines 95 to 142. The authors describe some very general characteristics of mobile genetic elements and the resistance genes with which they have been associated. It would be important to include other resistance mobility elements, such as integrons.
Lines 143 to 169. The authors describe some molecular methodologies for diagnosis but do not explain the basis for these statements. They also describe some examples of “a little bit of everything” but do not describe them with specificity.
Lines 170 – 248. The authors describe the importance of genetic markers but do not mention what the characteristics of each one . They mention bioinformatics methodologies and the existing problems, but they do not describe them in detail or explain them well.
Lines 249 – 255. From this perspective, they describe how making an accurate diagnosis in terms of antibiotic-resistance genes and virulence factors will revolutionize how infections are diagnosed and treated. However, they never mention how, what tools we have now, how they work, or how they can improve.
The references must be updated; only 21.6% (8/37) are from publications from the last five years.
Other comments
1. What is the central question addressed by the research?
They discuss the identification and treatment of bacterial infections in the molecular era and propose changing from an identification strategy based on traditional phenotypic taxonomy to one based on genetic identification.
2. What parts do you consider original or relevant for the field? What specific gap in the field does the paper address?
Considers the impact of whole genome sequencing on identifying species that require more time at a microbiological level. However, one of the problems that is important to highlight is that it does not impact the processing of data after sequencing. Consider detailing what knowledge is required to conduct subsequent data analysis since this part is only in a few lines in the text. Likewise, consider whether an MGE is the target for typing.
3. What does it add to the subject area compared with other published material?
The paper does not describe the molecular techniques; it only mentions two well-known methods but does not describe them.
4. What specific improvements should the authors consider regarding the methodology? What further controls should be considered?
They must improve by giving examples of the molecular/genetic methodologies they want to emphasize, describing them, and comparing them with those used nowadays.
5. Please describe how the conclusions are or are not consistent with the evidence and arguments presented. Please also indicate if all main questions were addressed and by which specific experiments.
The conclusions could be more consistent. The evidence and arguments presented need to describe how they work, how they can be improved, and the challenges.
6. Please include any additional comments on the tables and figures and quality of the data.
I invite the authors to improve their concise paper and explain the methodologies, molecular and genetic, beginning from the phenotypic (how does it work, and does it still work?).
They must develop their ideas, but they still need to be completed. Acronyms must first be defined as they appear in the text. Authors must include one or two tables highlighting relevant information in the manuscript. The authors must include a conclusions section. The authors must include a figure that supports the conclusions of the review or manuscript.
Comments on the Quality of English Language
The quality of the English is good, it only requires minimal revision.
Reviewer 2 Report
Comments and Suggestions for Authors
This manuscript highlights the necessity of transitioning from species identification to gene-based diagnostics, emphasizing that traditional phenotype-based classification methods are no longer adequate to address the challenges posed by modern pathogens. By adopting molecular technologies such as whole-genome sequencing, microbiologists can more accurately identify pathogens and trace the spread of antibiotic resistance genes. This new diagnostic paradigm not only improves diagnostic accuracy but also provides a foundation for more targeted antimicrobial therapy, aiding in the global fight against antibiotic resistance.
However, the review article is overall too general, and more detailed information should be included before publication.
1. Molecular diagnostics are not entirely absent today. Summarizing the current molecular diagnostic methods would help readers quickly understand the latest research and application trends. For example, while the authors mention whole-genome sequencing as a significant molecular diagnostic tool, other methods based on drug susceptibility testing or resistance gene detection (PCR, fluorescent in situ hybridization) and even protein-based detection methods (such as mass spectrometry) should also be discussed and compared.
2. What are the advantages of gene-based diagnostics? How do they compare to traditional methods? Issues such as detection cost and timeliness should be addressed.
3. Molecular diagnostics require high expertise and are often expensive. Additionally, antibiotic-resistant infections frequently occur in economically underdeveloped regions. How can these methods be promoted in such settings? Clinical testing methods should be reliable and reproducible, necessitating methodological maturity. What standardized analytical methods currently exist? What are the challenges in promoting molecular diagnostics?
In conclusion, while the author's review is forward-looking, more details should be included to substantiate the advantages of molecular diagnostics in diagnosing and treating infectious diseases before it can be accepted for publication.
Round 2
Reviewer 1 Report
Comments and Suggestions for Authors
All suggested comments have been made.
Reviewer 2 Report
Comments and Suggestions for Authors
After the author's revisions, the manuscript is more readable than before, and I have no further comments.